



**Title:** Comment on "Invariability of relationship between the polar cap magnetic activity and
geoeffective interplanetary electric field" by Troshichev et al. (2011).
**Author:** Peter Stauning, Danish Meteorological Institute, Lyngbyvej 100, Copenhagen.
Mail: pst@dmi.dk
**Abstract.** In the publication Troshichev et al. (2006) on the Polar Cap (PC) indices, PCN (North)
and PCS (South), an error was made by using components of the Interplanetary Magnetic Field
(IMF) in their Geocentric Solar Ecliptic (GSE) representation instead of the prescribed Geocentric
Solar Magnetosphere (GSM) representation for calculations of index scaling parameters in the
version AARI_1998-2001 (named AARI#3) issued from the Arctic and Antarctic Research Institute
(AARI) in St Petersburg, Russia. The mistake has caused a trail of incorrect relations and wrong
conclusions extending since 2006 up to now (2020). For the publication commented here,
Troshichev et al. (2011), the authors state that they have used scaling parameters of the (invalid)
AARI#3 PC index version in their work but they have actually substituted parameters from the
more recent AARI_1995-2005 (AARI#4) version instead. The mingling of PC index versions have
resulted in erroneous illustrations in their Figs. 1, 2, 3, 6, 7, and 8 and the issuing of non-
substantiated statements.

## 1. Introduction.

The publication Troshichev et al. (2006), hereinafter TJS2006, describes principles of a unified
calculation procedure using polar magnetic observations to derive values of Polar Cap (PC) indices
PCN (North) and PCS (South) agreed between the Arctic and Antarctic Research Institute (AARI)
in St. Petersburg and the Danish Meteorological Institute (DMI). PCN indices are based on
magnetic variations measured at Qaanaaq (THL) in Greenland while PCS indices are based on data
from Vostok in Antarctica.
The polar cap indices reflect the magnetic variations caused by the electric current systems (Hall
currents) associated with the transpolar convection of ionized plasma and embedded magnetic fields
driven by polar electric fields induced by solar wind - magnetosphere interactions. The magnetic
variations are scaled with respect to the merging electric field, $E_M$, in the impinging solar wind (Kan
and Lee, 1979) in order to make the index independent of local ionospheric properties, in particular,
the variable conductivities.
New analyses has disclosed that the use in TJS2006 of Interplanetary Magnetic Field (IMF)
components IMF $B_Y$ and IMF $B_Z$ in their Geocentric Solar Ecliptic (GSE) representation instead of
the prescribed Geocentric Solar Magnetosphere (GSM) representation have had grave consequences
for the PC index scaling parameters and index values. The GSE and GSM components of IMF
differ by a rotation around the common IMF $B_X$ direction by ±11.4° (magnetic dipole offset) in the
daily variation superimposed on the ±23.5° (eclipse angle) seasonal variation, that is, a total
variation of ±34.9° throughout the year. These varying differences have strong impacts on the
calculation of scaling parameters for the PC indices





The mistake is illustrated in Fig. 1 here where the IMF $B_Y$ and $B_Z$ components, displayed in
TJS2006 without mentioning of their reference system, are reproduced from Fig. 7 of Troshichev et
al. (2006) in Fig. 1a to be compared with their appearance in the GSE and GSM representation
displayed in Fig. 1b. The differences between the GSE and GSM versions are most easily distinguishable
between 12 and 14 UT where IMF $B_Z$(GSE) is positive while $B_Z$(GSM) is negative.

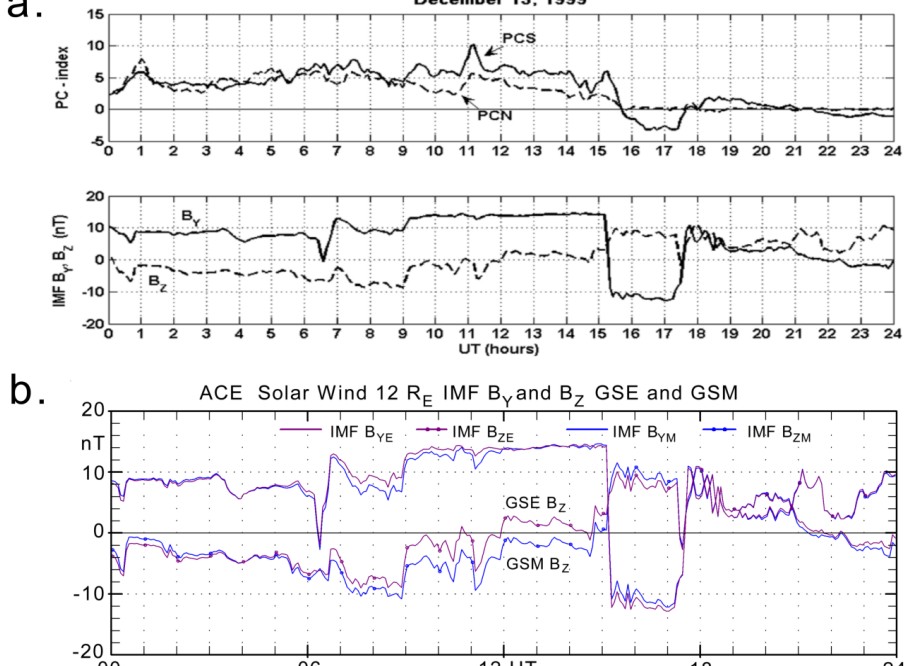


**Figure 1.** (**a**) IMF $B_Y$ and $B_Z$ components reproduced from Fig. 7 of Troshichev et al., 2006. (**b**) IMF $B_Y$ and
$B_Z$ components in their GSE version (magenta line) and in their GSM version (blue line). The differences
between GSE and GSM versions of IMF $B_Z$ are clearly discernible between 12 and 14 UT.

The mistake had no strong impact on the remaining presentation of the PC index concept in
TJS2006. Usually, such a mistake would not attract attention after the many years that have passed
since the publishing in 2006. However, the incorrect feature drags a trail of erroneous relations and
invalid statements presented in publications on polar cap indices issued since 2006 extending up to
present (2020).
Thus, the scaling parameter sets presented in the colour-coded diagrams of Figure 3 of TJS2006
have been reproduced in Troshichev et al. (2011), in Troshichev and Janzhura (2012), and in
Troshichev (2011) that all form part of the basis for the IAGA-recommended PC index versions
(Matzka, 2014; Nielsen and Willer, 2019). Most recently, the TJS2006 publication and the incorrect
results from the derived publication, Troshichev et al. (2011), have been referenced in Troshichev
(2017) and in the technical report, ISO/TR 23989: 2020, issued by the International Standards
Organization (ISO) in January 2020.




**2. Erroneous scaling parameters for the PCS indices**.
In the agreed formulation, the PC indices are derived from the expression shown in Eq. 1 here (see,
e.g., TJS2006; Stauning et al., 2006):
$\quad$ PC = $(\Delta F_{\mathrm{PROJ}} - \beta)/\alpha$ $\hfill$ (1)
where $\Delta F_{\mathrm{PROJ}}$ is the projection to an optimal direction of the horizontal magnetic disturbance vector
measured from a quiet reference level while $\alpha$ (slope) and $\beta$ (intercept) are calibration parameters.
With the magnetic components in their geographic (X,Y) representation and *UTh* the UT time in
hours, the projection angle is defined by Eq. 2:
$\quad$ $V_{\mathrm{PROJ}}$ = observatory longitude($\lambda$) + $UTh \cdot 15°$+ optimum direction angle($\varphi$) $\hfill$ (2)
The optimum direction is characterized by its angle ($\varphi$) with the dawn-dusk meridian and derived
from seeking optimal correlation between $\Delta F_{\mathrm{PROJ}}$ and the solar wind merging electric field, $E_{\mathrm{M}}$, in
the formulation of Kan and Lee (1979) based on using IMF components in their GSM
representation.
In Troshichev et al. (2006), the derived PCN and PCS scaling parameters ($\varphi$, $\alpha$, $\beta$) are presented in
the colour coded diagrams in their Fig. 3, which is reproduced here (including caption) in Fig. 2 for
convenience. This version from 2006 was named "AARI#3" by McCready and Menvielle (2010,
84  2011).

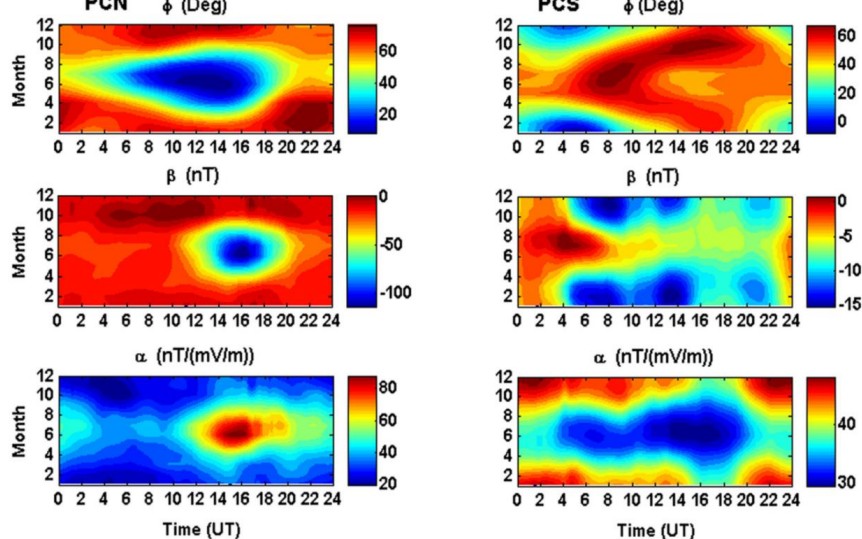

**Figure 3.** Angle $\phi$ and coefficients $\alpha$ and $\beta$ used for calculation of the unified PCN and PCS indices
derived on the basis of magnetic data from Thule and Vostok stations for 1998–2001.


**Fig. 2.** Reproduction of colour-coded displays of PC index scaling parameters from Fig. 3 of Troshichev et
al. (2006).

In coarse terms the IMF $B_{\mathrm{Z}}$ component mainly affects the noon-midnight flow intensity while the
IMF $B_{\mathrm{Y}}$ component mainly affects the dawn-dusk component of the transpolar flow of plasma and
embedded magnetic fields that generate the polar magnetic variations represented in the Polar Cap
(PC) indices,. Thus, the relation between the two IMF components affects the transpolar flow
intensity and, in particular, its direction. Consequently, the main effect of the different GSE/GSM





IMF representations would be found in the optimum direction assumed perpendicular to the
dominant flow direction.
In the derived publication, Troshichev, Podorozhkina, and Janzhura (2011) (hereinafter TPJ2011),
the colour-coded diagrams for PCS scaling parameters in version AARI_1998-2001 (AARI#3)
presented in the right column of Fig. 3 of TJS2006 (Fig. 2 here) are displayed in the left column of
their Fig. 5 (here reproduced including caption in Fig. 3). These values are considered to represent
PCS scaling parameters for a solar maximum epoch. The figure has also a column (left) for the
scaling parameters in the later version AARI_1995-2005 (AARI#4) based on data from the epoch
1995-2005 spanning an entire solar cycle. The middle column in their Fig. 5 (Fig. 3 here) presents
scaling parameters based on the solar minimum years 1997+2007-2009, here named version
AARI_1997+2007-2008 taken to represent solar minimum scaling parameters.

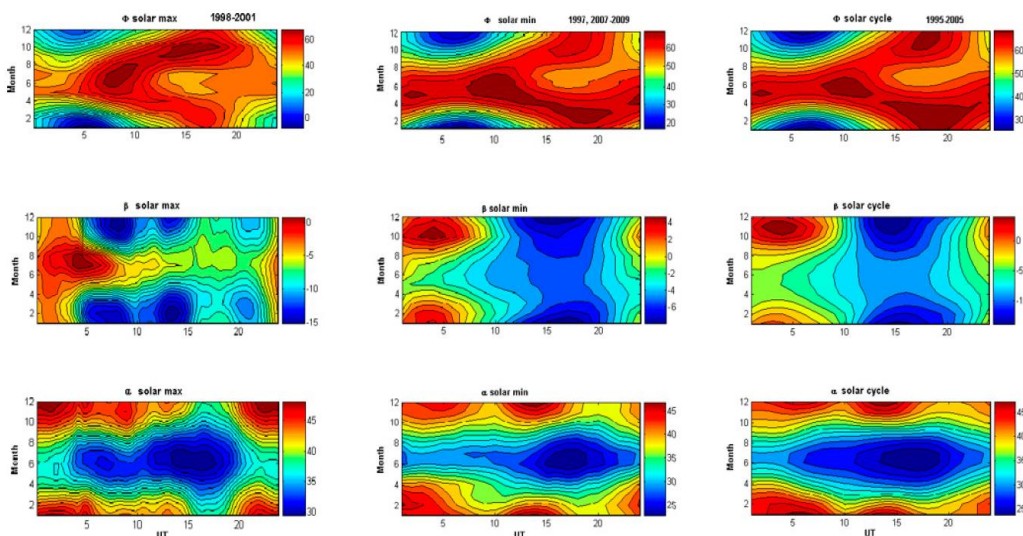

**Fig. 5.** Parameters $\phi$, $\beta$, and $\alpha$ derived for Vostok station independently for epoch of solar maximum (1998–2001) [Troshichev et al., 2006], for epoch of solar minimum (1997, 2007–2008), and for complete cycle of solar activity (1995–2005) (AARI#3 version); the axis of abscises being for UT and axis of ordinates being for month.

**Fig. 3.** PCS scaling parameters in colour-coded formats for (left) solar max. version AARI_1998-2001,
(middle) solar minimum version AARI_1997+2007-2009, and (right) average solar cycle version
AARI_1995-2005. (Reproduced from Troshichev et al., 2011, note error 2008 instead of 2009 in caption).

A problem for the analysis of possible effects of the invalid PCS scaling parameters derived in
TJS2006 by using IMF components in their GSE representation is the unavailability of numerical
files of the parameters.
Instead, the colour-coded diagrams have been "manually" read-off to be converted to numerical
files. Actually, the readings of PCS scaling parameters from the right column of Fig. 3 of TJS2006
(Fig. 2 here) have been consolidated by the readings of the corresponding diagrams in Fig. 5 of
TPJ2011 (Fig. 3 here) where the colour coding has been supplemented by contour curves, which
facilitates the reading of values. Using the colour coded scales to the right of each diagram, the
parameter values have been read-off and converted from the graphical representation into the files
of mean hourly values shown in Table 1 of the appendix.





For the full cycle (1995-2005) the scaling parameters in version AARI_1995-2005 (AARI#4) have
been provided in files (Angle_Fi.1M, Coeff_alpha.1M, Coeff_beta.1M) supplied from AARI at an
earlier communication ("Parameter.rar", Janzhura 21-06-2011). The mean hourly values derived
from these files are shown in Table 2 of the appendix.
The investigations reported in their Figs. 6, 7, and 8 seem to indicate that the PCS index values
derived by using the "solar max" parameters of the AARI#3 version from 2006 are very close (in
p.1488 declared to be "*within 10%*") to the PCS values derived with the "solar min" scaling
parameters in the AARI_1997+2007-2009 version. Thus, it is concluded in TPJ2011 that scaling
parameters derived using appropriate quiet day reference (QDC) handling are virtually independent
of the solar cycle.
However, by some further mistake, the AARI#3 scaling parameters in version AARI_1998-2001
from TJS2006 are not at all used in the reported examinations. It appears that the scaling parameters
from version AARI_1995-2005 (AARI#4) have been inserted (without mentioning) to substitute for
the (erroneous) parameters of version AARI_1998-2001 (AARI#3) in the QDC analyses related to
their Figs. 1, 2, and 3. It has not been possible to deduce the origin of the scaling parameters
actually used for two PCS versions being compared in Figs. 6, 7, and 8 of TPJ2011.
Thus, it appears that the TPJ2011 publication fails to recognize the problems with the adverse
scaling parameters in the version AARI_1998-2001 (AARI#3), which have been used by the
authors for further publications throughout some years since it was developed in 2006. By stating to
use version AARI#3 scaling parameters for calculations of PC indices and then demonstrate the
small differences between PC index values derived by tacitly using scaling parameters of two
slightly different AARI#4 versions, they avoid to demonstrate the failure of the AARI_1998-2001
(AARI#3) version from 2006. Instead, in the caption of their Fig. 5 (Fig.3 here) the authors make
version AARI_1995-2005 become "AARI#3" which make the real AARI#3 version from 2006
vanish.
The substitute of versions is supported by incorrect quotations. In p. 1479 of TPJ2011 the authors
write: "*The parameters α, β, φ derived for full cycle of solar activity (1995-2005) were used in the*
*procedure adopted in the Arctic and Antarctic Research Institute for the unified PC index*
*derivation (the procedure known as AARI#3 version, according to the nomenclature proposed by*
*McCready and Menvielle, 2010).*"
However, McCready and Menvielle (2010) note in their summary Table 1 (pp. 1888-1889) on the
different PC index versions that: "*AARI#3_2006, ACE 1998-2001, official PCS index*".
The mingling of scaling parameter versions in the discussion of the effects of using quiet day levels
(QDC) or just base levels (without QDC) in the reference levels used for processing of magnetic
data to derive PC index values has generated obviously incorrect results easily spotted in the
"optimum angle" and "slope" scaling parameters displayed in their Fig. 1 as demonstrated in
section 3 here.


**3. Examination of the PCS scaling parameters used in Troshichev et al. (2011).**
This section examines in detail the use of PCS scaling parameters in TPJ2011. One line of
examinations concern identification of the PCS version actually used in the analyses. The other line
of examinations regards the validity of the reported results assuming that the PCS version
substituted for the erroneous AARI_1998-2001 version (AARI#3) has adequate properties.





### 3.1. Identification of the PCS parameter version

The QDC issue is the question whether the polar magnetic variations used in Eq. 1 should be measured from the secularly varying base level or from the varying level (QDC) recorded during "*extremely quiescent days*" (TJS2006). (see Janzhura and Troshichev, 2008, for details)

Fig. 1 of TPJ2011 is meant to provide basis for a discussion of the importance of using QDC correction of the reference level for observed magnetic data at calculations of PC index scaling parameter and index values. The diagrams of their Figs. 1a, b, and c display daily variations in the optimum angle, φ, the slope of the regression line, α, and the intercept, β, derived without using QDC (thin blue lines) and with use of QDC (thick green lines) for the same local winter (15 June) and summer (15 November) days.

There are two essential problems with their Fig. 1. Against their statements, the "with QDC" curves are not derived as stated from the AARI_1998-2001 (AARI#3) version from TJS2006. They are taken from the more recent AARI_1995-2005 (AARI#4) scaling parameter version. Furthermore, the "without QDC" curves are not derived from calculations of scaling parameters from the "with QDC" version just without using QDCs but are of indefinable origin.

Figs. 4a,b here displays in green line the optimum angles read from the "with QDC" curves in Fig. 1a. The angle values derived from the parameter file, Angle_Fi.1M, derived for epoch 1995-2005 are displayed in blue dashed line, and the corresponding angles read from the left column (epoch 1998-2001) of their Fig. 5 (Fig.3 here) are displayed by the red line with dots.

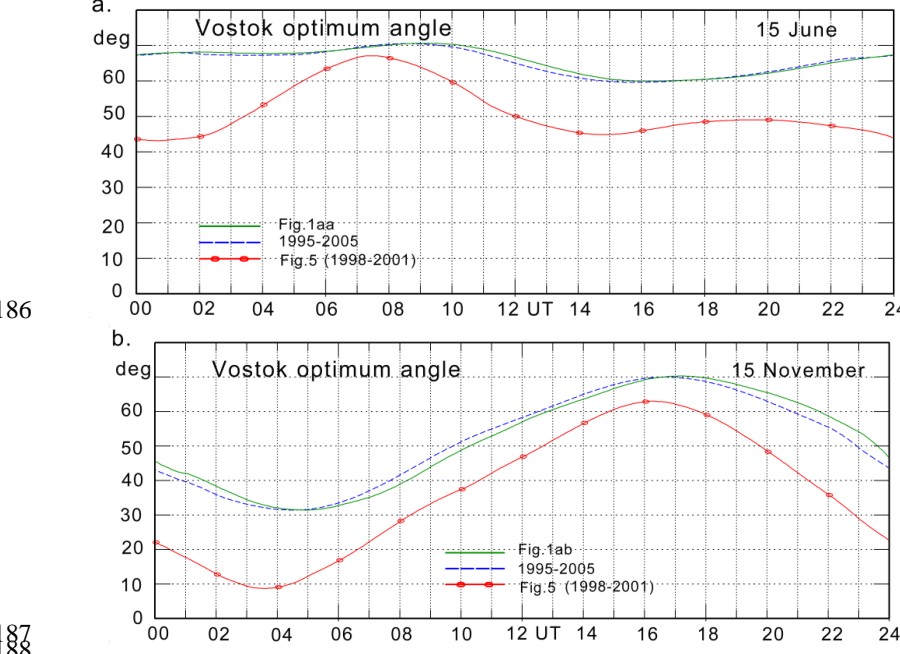

**Fig. 4.** (**a**) Vostok optimum angles on 15 June. Angles read from Fig. 1aa of Troshichev et al., 2011, in green line. Angles from AARI file (Coeff_Fi.1M, 21-06-2011), epoch 1995-2005, in blue, dashed line. Angles read from the left column of their Fig. 5 (epoch 1998-2001) in red line with dots. (**b**) The corresponding diagram for 15 November (Fig.1ab) using notation and line colours like those of Fig. 4a.



From the displays of optimum angles by the green lines in Figs. 4a and 4b here it is evident that the
angles represented by the "with QDC" solid green lines in Fig. 1a of TPJ2011 for 15 June and 15
November represent the AARI_1995-2005 version presented in Fig. 4 here in blue, dashed line, and
not the AARI_1998-2001 version (derived by Troshichev et al., 2006) represented here by the red
line with dots. The optimum angles from the AARI_1998-2001 (AARI#3) version (red line, dots)
differ by up to 25° (in June) from the other two optimum angle versions.
Concerning the PCS slope (α) coefficients, Figs. 5a,b here displays in green line the slope values
displayed by the "with QDC" heavy green line in Fig. 1b (15 June and 15 November) of TPJ2011.
The slope values defined in the AARI file Coeff_alpha.1M (21-06-2011) (epoch 1995-2005) are
displayed in dashed blue line while the slope values from the AARI_1998-2001 version read from
the left column of their Fig. 5 are displayed by the red line with dots.

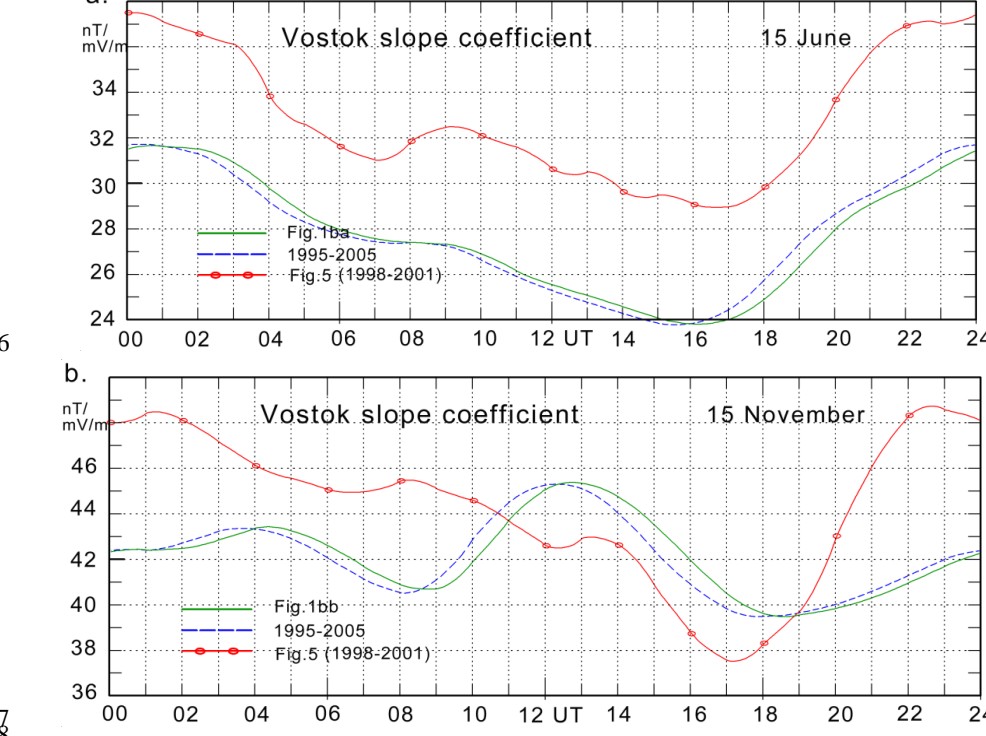

**Fig. 5.** (**a**) Vostok PCS slope coefficients 15 June (with QDC). Slope values read from Fig. 1ba of
Troshichev et al., 2011 in green line. Slope values from AARI file (Coeff_alpha.1M, 21-06-2011), epoch
1995-2005, in blue dashed line. Slope values read from left column of their Fig. 5 (epoch 1998-2001) in red
line with dots. (**b**) The corresponding diagram for 15 November (ref. Fig.1bb) using notation and line colours
like those of Fig. 5a.

Again, like inferred from the displays of optimum angles, the "with-QDC" curves in heavy green
lines in Fig. 1b of Troshichev et al. (2011) represent slope values from the AARI_1995-2005
version (AARI#4) and not the AARI_1998-2001 version (AARI#3) from Troshichev et al. (2006).





Concerning identification of the version used in TPJ2011 for the intercept (β) values in the
diagrams displayed in their Fig. 1c, the "with QDC" curves (in heavy green line) provide again, as
seen in Figs. 6a,b here, values derived from the AARI_1995-2005 version (AARI#4) and not the
AARI_1998-2001 version (AARI#3) as claimed in their statements.

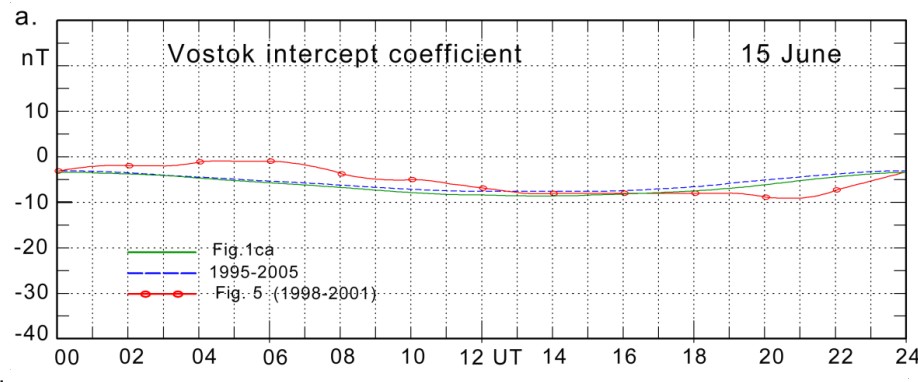


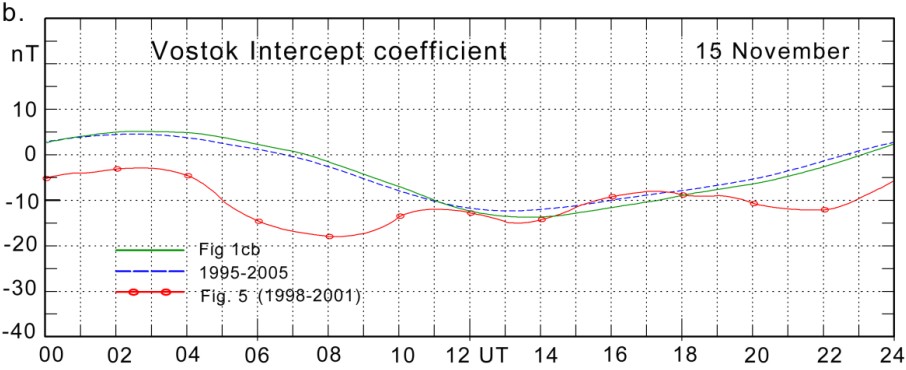


**Fig. 6** (**a**) Vostok intercept coefficients 15 June (with QDC). Intercept values read from Fig. 1ca of
Troshichev et al., 2011, in green line. Intercept values from AARI file (Coeff_beta.1M, 21-06-2011),
epoch 1995-2005, in blue dashed line. Intercept values read from left column of their Fig. 5 (epoch 1998-
2001) in red line with dots. (**b**) The corresponding diagram for 15 November (ref. Fig.1cb) using notation
and line colours like those of Fig. 6a.

The close correspondence between values in the AARI files of scaling parameters derived for epoch
1995-2005 version (AARI#4) and the values read from the "with QDC" curves in Figs. 1a, b, c
leaves no doubt that they are derived from the same scaling parameter version. In spite of possible
inaccuracies in the reading of values from the colour-coded diagrams it is clear that the values
represented by the red curves with dots in Figs. 4b, 5b, and 6b here are not displayed in Fig. 1 of
TPJ2011. Thus, the statement in p. 1484 of TPJ2011, claiming that the scaling parameter values
shown in their Fig. 5 based on epoch 1998-2001 have been used for the displays in their Fig. 1, is
incorrect.

**3.2. The QDC effects on PCS scaling parameters**



Like noted above, the text to the diagrams in Figs. 1a, b, and c of TPJ2011 claim they display daily
variations in the optimum angle, φ, the slope of the regression line, α, and the regression intercept,
β, derived without using QDC (thin blue lines) and with use of QDC (thick green lines) for the same
local winter (15 June) and summer (15 November) days.
In p. 1484 the authors write: "*To demonstrate the QDC role in derivation of α, β, and φ parameters,*
*the parameters derived with inclusion of the QDC and without QDC should be compared. To*
*provide such comparison, in our analysis we used the same experimental data (Satellite*
*measurements of EKL and magnetic data from Vostok for 1998-2001) to derive a set of parameters*
*$\alpha_0$, $\beta_0$, and $\varphi_0$ without including the QDC. Results of this calculation – angle $\varphi_0$, slope of regression*
*$\beta_0$ and intersection $\beta_0$ - are shown in Fig. 1 for winter and summer days at the Vostok station (15*
*June and 15 November 2002, respectively) along with parameters φ, α, and β derived for the same*
*days with inclusion of QDC.*"
For the data displayed in heavy green line in their Fig. 1a reproduced from Troshichev et al. (2011)
in Fig. 7a here, it is stated in p. 1484 of TPJ2011, as quoted above, that they present PCS optimum
angles derived from magnetic data from Vostok for 1998-2001 with QDC corrections of the
reference levels. For the data displayed in thin blue line in their Fig. 1a reproduced in Fig. 7a, it is
stated that they present PCS optimum angles derived from the same data but without using QDC
reference level corrections.
However, it is seen at a glance that this could not be correct. Optimum angle values are derived by
searching for optimum correlation between the merging electric field, $E_M$, (also denoted $E_{KL}$) in the
solar wind and the projected value of the horizontal polar magnetic disturbance vector. The QDC
represent the undisturbed variations on "*extremely quiescent days*" (quote from TJS2006) where $E_M$
≈ 0 and could not possibly affect the correlation of $\Delta F_{PROJ}$ with $E_M$ much. Thus, the optimum
angles with QDC and without QDC should be (almost) the same. It has not been possible to obtain
information on the real origin of the "without QDC" curves or to deduce their derivation by
examining available data.
PCS scaling parameters have been derived with a DMI program (Stauning et al., 2006; Stauning,
2016) where the QDC involvement can be switched in and out without affecting other steps in the
calculations. Another feature in the program is the possible adjustment of the averaging/smoothing
of the derived optimum angles. For the example for 15 November, Fig. 7b (middle field) here
presents the resulting optimum angles for the with/without-QDC cases for a light level of
smoothing. Fig. 7c (bottom field) presents the optimum angles for the QDC/no QDC cases with a
stronger level of averaging/smoothing. The differences between the re-calculated "with QDC" and
"without QDC" values are very small in both cases.

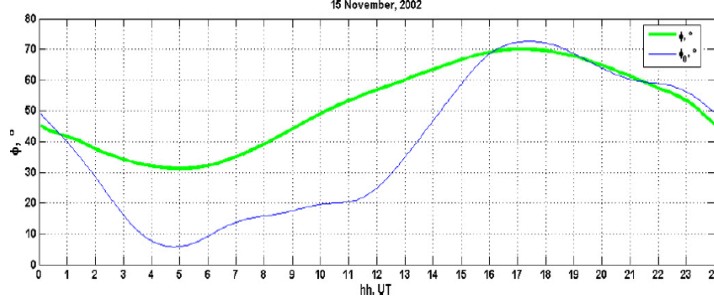




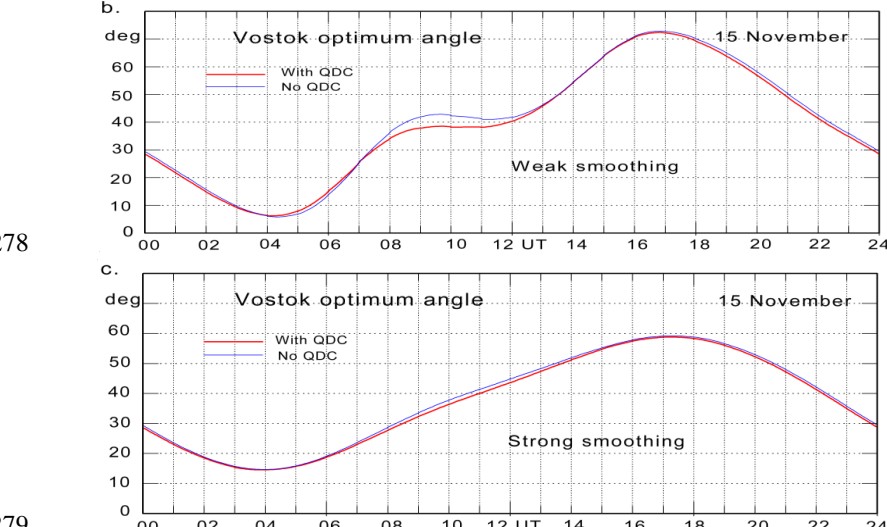

**Fig. 7.** Optimum angles for Vostok on 15 Nov. The top field (**a**) displays the with-QDC (heavy green line) and no-QDC (thin blue line) calculations of optimum angles by Troshichev et al. (2011) reproduced from their Fig. 1ab. Middle field (**b**) displays results from the re-calculation with and without QDC with light smoothing. Bottom field (**c**) displays the re-calculation of optimum angles with and without QDC with strong averaging/smoothing.

The slope values (α) for the "with QDC" and "without QDC" cases should also be nearly the same since the samples of magnetic disturbance data used for the regression line are all displaced (parallel-shifted) by the same QDC-related amount. The intercept values (β) will change by this amount (see Stauning, 2013). The relations between slope values in Fig. 1bb of TPJ2011 and re-calculated values are displayed in Fig. 8 while the relations between intercept values in Fig. 1cb of TPJ2011 and recalculated values are shown in Fig, 9

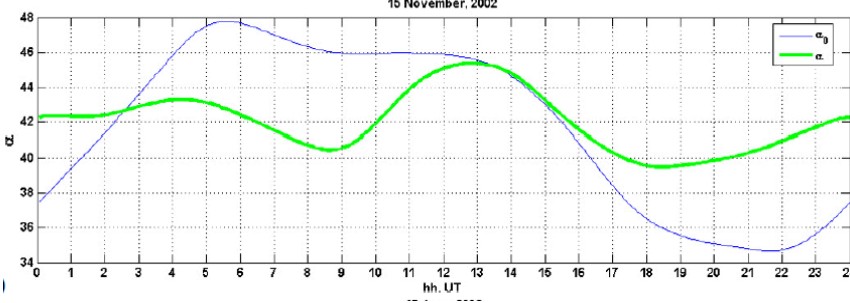





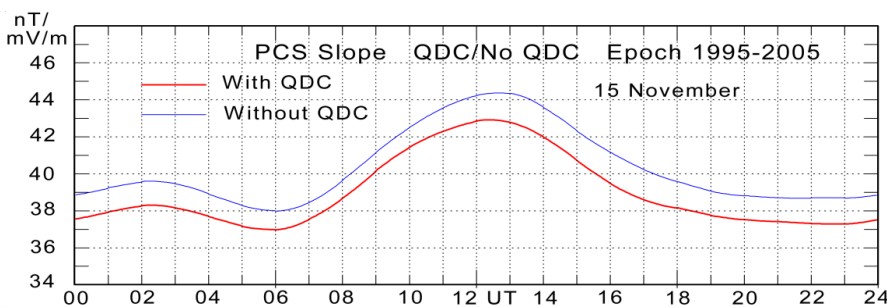

**Fig. 8.** Display of slope values, α, for 15 Nov to be used for derivation of PCS indices. Top field: slope values reproduced from Fig. 1bb of Troshichev et al., 2011. Bottom: re-calculation of slopes with QDC (red) and without QDC (blue)

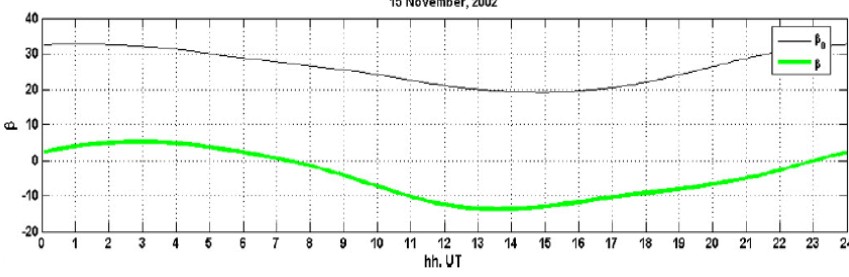

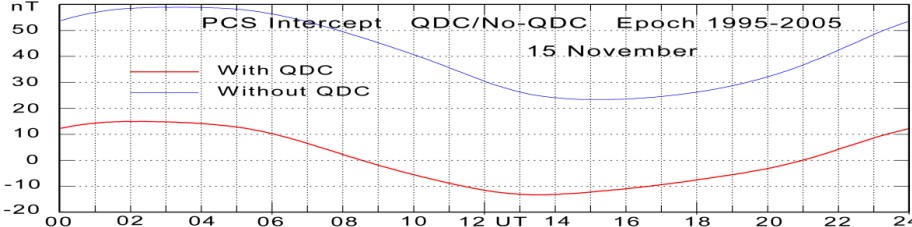

**Fig. 9.** Top field: PCS intercept values 15 Nov 2001 reproduced from Fig. 1cb of Troshichev et al., 2011. Intercept values derived with QDC in heavy green line. Without QDC in thin blue line. Bottom: Display of intercept values, β, for 15 Nov calculated with QDC (red) and without QDC (blue).

### 3.3. PCS values calculated with/without QDC.

Re-calculated values of the with-QDC/no-QDC coefficient sets α, β, and φ have been used to re-calculate PCS index values with and without QDC reduction of Vostok geomagnetic data. The re-calculated PCS values corresponding to those of Figs. 2a and 2b of TPJ2011 are displayed in Fig. 10.

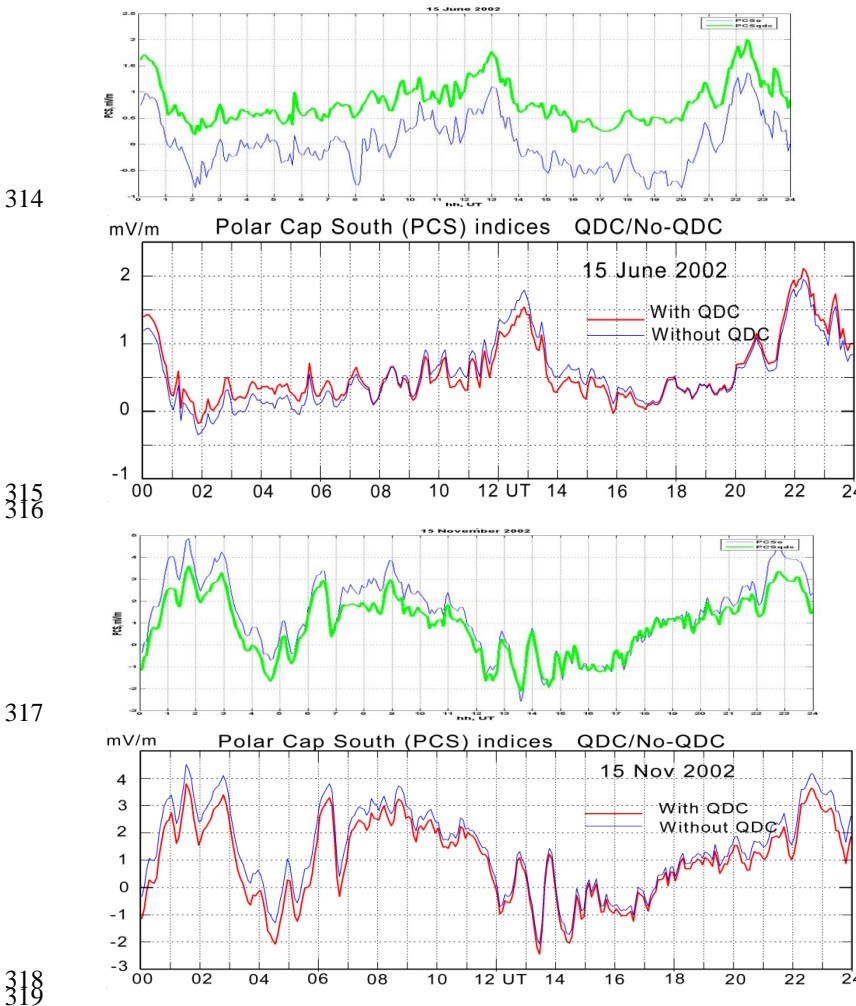

**Fig. 10.** PCS indices calculated with/without QDC. Top field: PCS index values for 15 June 2002 reproduced from Fig. 2a of Troshichev et al. (2011). Next lower field: Recalculation for 15 June 2002. Lower two fields present corresponding sets for 15 November 2002.

It is evident from the examples in Fig. 10 that the differences between the "with QDC" and the "without QDC" cases have been substantially reduced in the re-calculations. Actually, an epoch-average QDC correction is built into the intercept (β) scaling parameter as explained in Stauning (2013). When the same "with/without QDC" procedure is used for calculation of the scaling parameters as well as for the calculation of PC index values then the differences are rather small.

The overall results for 2002 are displayed in the bottom field of Fig. 11 here in the format of Fig. 3 from TPJ2011.




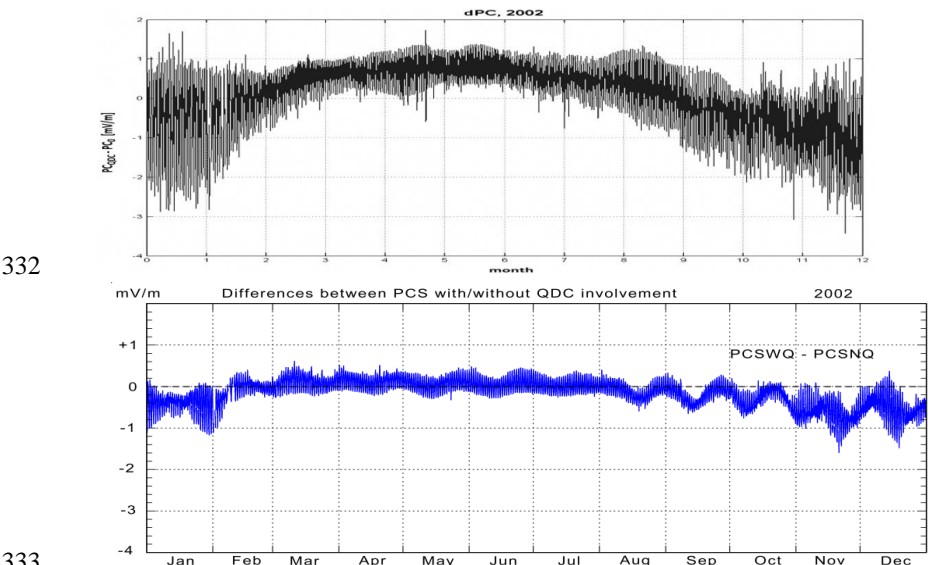


**Fig. 11**. Display of differences between PCS values calculated with and without QDC reductions of Vostok
magnetic data for 2002. Top field: Reproduced from Fig. 3 of Troshichev et al., 2011. Bottom: Re-
calculation of the PCS with-QDC/no-QDC index differences.

The top field of Fig. 11 presents the differences between the with-QDC/no-QDC PCS index values
throughout 2002 displayed in Fig. 3, p.1483, of TPJ2011, while the diagram in the bottom field of
Fig. 11 presents the corresponding re-calculated values using reference levels with and without
QDC reduction. The plots in Figs. 10 and 11 indicate that the differences between PCS index values
calculated with and without QDC reduction of Vostok data are 2-3 times larger in TPJ2011
publication than in the re-calculations. The main reason for the enhanced differences in the
TPJ2011 version is the introduction of an incorrect "without QDC" scaling parameter version (of
unknown origin) shown by the thin blue lines in their Fig. 1.

**3.4. Differences in PC index values for different sets of scaling parameters.**
According to the statements in TPJ2011, the PCS values and their differences displayed in Figs. 6,
7, and 8 have been derived from using the "solar max" scaling parameters (AARI_1998-2001)
displayed in Fig. 3 of TJS2006 (or Fig. 5 of TPJ2011) and the "solar min" scaling parameters in
version AARI_1997+2007-09 displayed in the middle column of the diagrams in their Fig. 5. The
"solar max" and "solar min" PCS values are superimposed on each other in the top fields of Fig. 6
(December 2001), Fig. 7 (June 2001), and Fig. 8 (year 2001). Their current differences are
displayed in the middle fields while the bottom fields display statistics on the distribution of
difference samples. Fig 12a here displays the TPJ2011 results for December 2001. It is seen from
the display of the statistics that the overwhelming majority of events are constrained within ± 0.2
mV/m.
With the index scaling parameters read from Fig. 5 of TPJ2011 for versions AARI_1998-2001
(AARI#3, solar max) and AARI_1997+2007-09 (solar min) and using Vostok magnetic data
supplied from INTERMAGNET, the corresponding PCS index values have been calculated for the
same cases. The results for December 2001 are displayed in Fig. 12b in the format of Fig. 12a.





Now, the corresponding majority of events are held within about ± 1 mV/m with differences
ranging up to 3.5 mV/m.

**a.**

**b.**



**Fig. 12.** Display of differences between PCS index values for December 2001 calculated with epoch 1998-
2001 scaling parameters and with epoch 1997+2007-2009 scaling parameters, respectively. (**a**) Reproduction
of Fig. 6a from Troshichev et al. (2011). (**b**) Re-calculations using readings of scaling parameters from Fig. 5
of TPJ2011.

The scaling parameters for the displays in Fig. 12b were derived from readings of the erroneous
version AARI_1998-2001 (AARI#3) from 2006. In order to see whether using proper versions
throughout could give small differences like those of Fig. 12a, the PCS indices derived from using
version AARI_1995-2005 (solar cycle) and AARI_1997+2007-09 (solar min) have been compared.
The results are displayed in Fig. 13.
The differences derived for this case are smaller than those presented in the recalculations based on
version AARI_1998-2001 (solar max) vs. version AARI_1997+2007-2009 (solar min) displayed in
Fig. 12b. They are still considerably larger than the differences displayed in Fig. 6 of TPJ2011 (Fig.
12a here) although, in principle, spanning only half the range between max and min solar activity,
they should be lower. Thus, the small differences displayed by the comparisons in Figs. 6, 7, and 8
are not based on PCS index values calculated with scaling parameters derived from different epochs
among those in play.


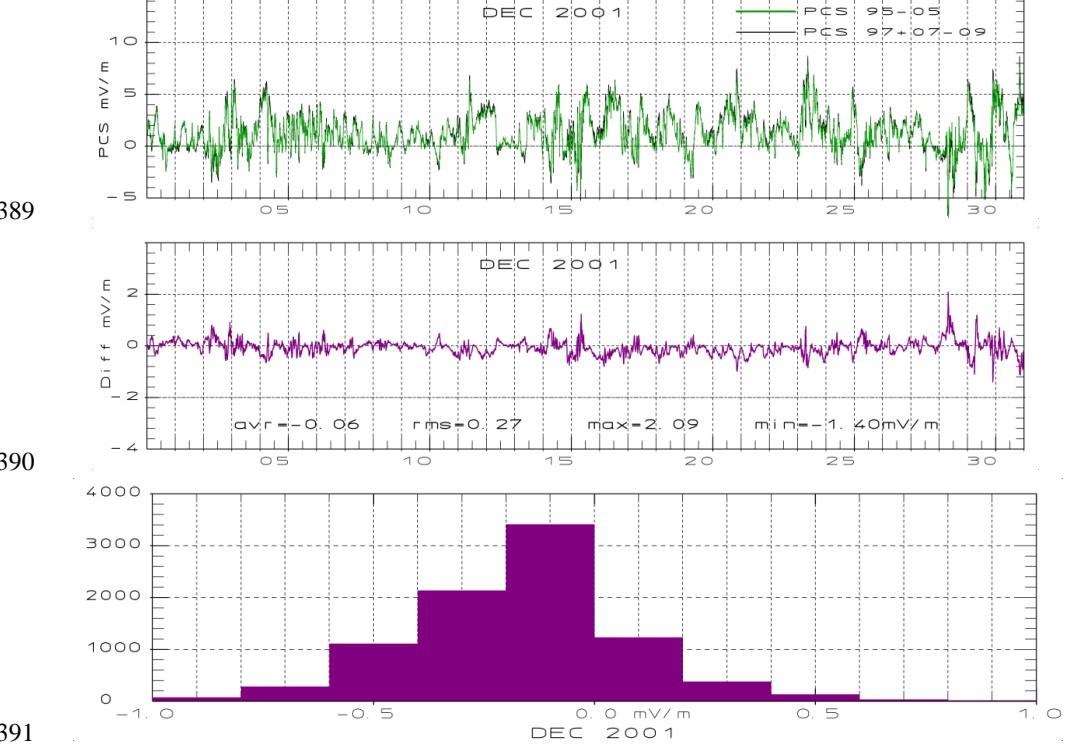

**Figure 13**. Calculation of PCS index values for December 2001 in versions 1995-2005 and 1997+2007-2009
and their differences in the format of Fig. 6 of Troshichev et al. (2011).





It has not been possible to deduce the origin of the scaling parameter sets used for calculations of the PCS
index values presented in Figs. 6, 7, and 8 in TPJ2011. However, it is evident that the authors have not used
the scaling parameters provided by the AARI#3 version from TJS2006.
The authors of TPJ2011 conclude (p. 1488) from their Figs. 6, 7, an 8 that the close consistency
between PC indices calculated with scaling parameters derived from epochs of high solar activity
(AARI_1998-2001) and from epochs of low solar activity (AARI_1997+2007-2009) indicates that
the scaling parameters "*can be considered as invariant with respect to solar activity*". However,
their conclusion rests on the erroneous substitute of another set of scaling parameters (presently not
known) for the solar maximum-based AARI_1998-2001 (AARI#3) scaling parameter set derived on
basis of the Troshichev et al. (2006) mistake in using IMF parameters in their GSE representation.
Thus, the conclusion in TPJ2011 is not properly substantiated.


**4. Differences in optimum angles and resulting PCS values between versions AARI#3 and #4.**
Results from the double reading of the PCS scaling coefficients for the optimum angle (φ) from Fig.
3 of TJS2006 and Fig. 5 of TPJ2011 are displayed by the green and red curves in Fig. 14 here. The
magenta curves in Fig. 14 presents PCS optimum angle values for version AARI_1995-2005
(AARI#4) provided in the file ("Parameter.rar", Janzhura 21-06-2011) from AARI.

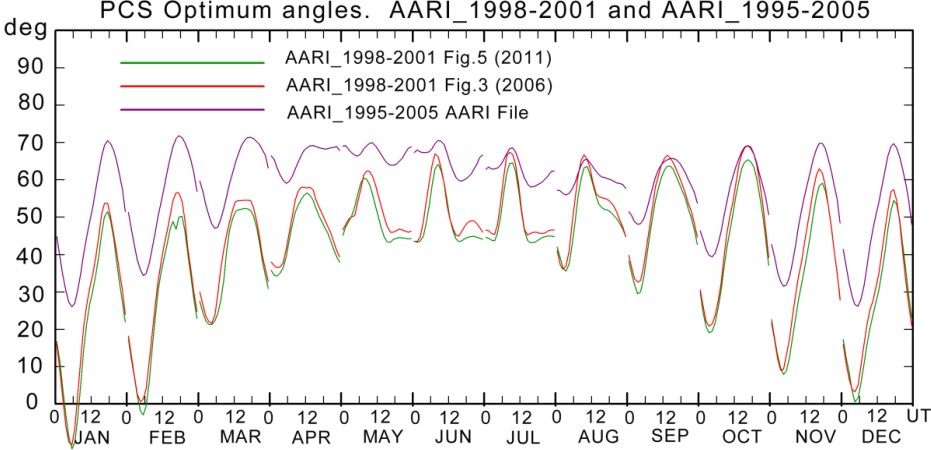


**Fig. 14.** Reading of the optimum angles for the PCS coefficients in version AARI_1998-2001 (AARI#3)
from the upper left diagram in Fig. 5a of Troshichev et al. (2011) in green line and those from upper right
diagram of Fig. 3 from Troshichev et al. (2006) in red line. Optimum angles in a numerical file for the PCS
version AARI_1995-2005 (Angle_Fi.1M ) are displayed by the uppermost magenta line.

For each of the 12 monthly sections of Fig. 14, the displayed curves present the monthly average
daily variation from 00 to 24 UT. The differences between optimum angles in the AARI_1998-2001
(AARI#3) and the AARI_1995-2005 (AARI#4) versions vary with time of the day and season
between 0° at appr. 10 UT in the southern winter season and up to almost 40° at appr. 06 UT in the
southern summer season. These variations in the differences are coupled to the systematic variations
in the angular differences between IMF components in the GSE vs. GSM representations.
The slope (α) and intercept (β) calibration parameters are also affected by the erroneous use of IMF
components in the GSE representation in TJS2006. When applied to calculations of PC indices



there are considerable differences between results derived from using the AARI_1998-2001 GSE-
based (AARI#3) and the AARI_1995-2005 GSM-based (AARI#4) scaling parameter versions. An
example of differences in the PCS calculations throughout 2001 is presented in Fig. 15.

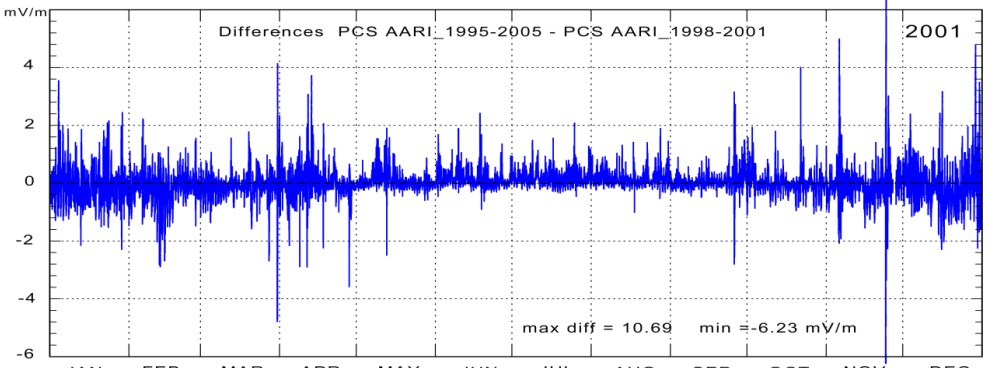

**Fig. 15**. Differences between PCS values derived with solar cycle average scaling parameters in the
AARI_1995-2005 (AARI#4) GSM-based version and PCS values derived with GSE-based scaling
parameters in the AARI_1998-2001 (AARI#3) version.

Generally, the differences range between ±1 mV/m during quiet or weakly disturbed conditions, but may rise
to range between ±2 mV/m during intervals of disturbed conditions. During magnetic storm events the
differences could be much larger to reach values in excess of 10 mV/m like noted in Fig. 15.
The erroneous PC index values might have affected individual cases used, for instance, in reported
magnetic storm or substorm investigations. It should also be noted that the systematic nature of the
errors in the PC indices related to systematic variations in the GSE vs. GSM transformations is
expected to invalidate statistical investigations based on using PC indices derived with the
erroneous scaling parameters in version AARI#3 resulting from the use of GSE-based IMF
components in TJS2006.

**5. Discussions**
In a natural and acceptable development, geomagnetic indices may change as new basic data arrive
or when the calculation methods are refined. Such changes should be revealed in the updated
documentation. However, changes resulting from detection of errors in the calculations should also
be reported to the scientific community. There is no question that the mistake in using GSE rather
than GSM representation in Troshichev et al., 2006 (TJS2006) is an error that has resulted in
incorrect values of the scaling parameters (φ, α, β) in the AARI_1998-2001 (AARI#3) version. The
error was detected in 2009 but at that time considered of minor importance. The grave
consequences of the mistake were not disclosed until the recent examination of the publication
Troshichev et al., 2011 (TPJ2011).
The stated main purpose of TPJ2011 was to demonstrate the invariability of PC index scaling
parameters derived on basis of data from epochs of high and low solar activity, respectively. A
secondary mission was to prove that including specifically calculated quiet day values (QDCs) in
the reference level was mandatory for obtaining proper PC index values. For both cases, reference
was made to the work presented in TJS2006 which included calculation of PCS index scaling





parameters for version AARI_1998-2001 (AARI#3) displayed in their Fig. 5 in a copy of the right
column of Fig. 3 of TJS2006.
However, in their Figs. 1, 2, and 3, against their statements, the scaling parameters in version
AARI_1995-2005 (AARI#4) and not the version AARI1998-2001 (AARI#3) were used for the
"with QDC" version, while the "without QDC" version displayed in their Fig. 1 and used for the
results in Figs. 2 and 3 is of unknown origin. The "without QDC" versions are definitely not
presenting scaling parameters obtained by just omitting the QDC involvement.
For their Figs. 6, 7, and 8 the authors state (p. 1486): "*To emphasize any differences in the*
*behaviour of parameters α, β, and φ in course of solar maximum and minimum epochs, the*
*coefficients presented in the left and middle columns of Fig. 5* (i.e., AARI_1998-2001 and
AARI_1997+2007-2009, respectively) *have been applied to calculate the appropriate values*
*($PC_{solmax}$) and $PC_{solmin}$) for the same year 2001.*" The small differences were taken to support their
conclusion that "*once derived parameters of α, β, and φ can be regarded as valid forever, provided*
*that the appropriate QDCs are used*"..
In both cases the authors, against their statements, fail to use the AARI_1998-2001 (AARI#3)
scaling parameters derived by Troshichev et al. (2006). Thus, their Figs. 1, 2, 3 and 6, 7, and 8 are
incorrect. It should be stressed that this judgement is not a matter of different opinions but a
conclusion drawn from documented errors in TJS2006 and TPJ2011.

**6. Importance of the commented publication, Troshichev, Podorozhkina, and Janzhura (2011)**
The mistake in the use of IMF components in their GSE instead of GSM representation has no
strong impact on the remaining presentation of the PC index concept in TJS2006. Usually, such a
mistake would not attract attention after the many years that have passed since the publishing in
2006. However, the incorrect features drag a trail of erroneous relations and invalid statements
presented in publications on polar cap indices issued since 2006 extending up to now (2020), among
others, in the commented publication: Troshichev, O. A., Podorozhkina, N. A., and Janzhura, A. S.:
Invariability of relationship between the polar cap magnetic activity and geoeffective interplanetary
electric field, Ann. Geophys., 29, 1479-1489, 2011.
Thus, the enlargement of the differences between PC index values derived with QDC vs. those
derived without QDC involvement presented in Figs. 2a, b and Fig. 3 of TPJ2011 may deter Space
Weather services from using the simple and reliable calculation of PC index values directly from
the magnetic variations with respect to the secular baseline using "without-QDC" scaling
parameters and accepting the implied small inaccuracies (less than ~1 mV/m).
The apparent small differences between the AARI_1998-2001 (AARI#3) and the AARI_1995-2005
(AARI#4) versions resulting from the hidden substitution of AARI#4 parameters might disguise
inaccuracies in further publications that have used the erroneous AARI#3 version.
The publications Troshichev et al. (2006) and Troshichev et al. (2011) have affected the
endorsement by IAGA of the present PC index versions. Basis for the endorsement is provided by
the "*Criteria for endorsement of indices by IAGA*" adopted in 2009. Here, section 2 reads:
"2. *The derivation of the index will be clearly defined; the algorithm will be available through*
*appropriate refereed and citeable publication(s); the algorithm must be shown to be independently*
*reproducible.*"
The material submitted from DTU Space and AARI to IAGA in 2013 in the application for
endorsement of their PC index versions (Matzka, 2014) states:





*"Regarding criterion 2:*
*The derivation of the index is described in the following publications:*
*Troshichev et al. (2006)*
*Janzhura and Troshichev (2008)*
*Janzhura and Troshichev (2011)*
*Troshichev and Janzhura (2012) (here, chapter 4 describes derivation of the provisional data set)"*

A replica of the TPJ2011 publication discussed here is included in chapter 4 of Troshichev and
Janzhura (2012) which form part of the abovementioned basis for the IAGA-endorsed index
calculation methodology (Matzka, 2014). Thus, the TPJ2011 publication along with the Troshichev
et al. (2006) and Janzhura and Troshichev (2011) publications have contributed to form basis for the
endorsement of the PC index versions by IAGA resolution #3 (2013).
Furthermore, the TPJ2011 publication and its non-substantiated results are referenced in p. 15 of
Troshichev (2017) and further referenced in page 4 of the ISO/TR23989:2020 technical report from
January 2020. The report is issued by the Technical Committee of the International Standards
Organization (ISO) considered to represent the ultimate authority in matters of the Space
Environment.


**Conclusions**
- The reported investigations in Troshichev, O. A., Podorozhkina, N. A., and Janzhura, A. S.
(2011): Invariability of relationship between the polar cap magnetic activity and geoeffective
interplanetary electric field, Ann. Geophys., 29, 1479-1489 (TPJ2011) are according to the authors
based, to a large extent, on the PC index version, AARI_1998-2001 (AARI#3), developed by
Troshichev et al. (2006). However, the AARI#3 version based on using IMF components in their
GSE instead of GSM representation is invalid and has generated odd scaling parameters.
- It appears that the TPJ2011 publication serves to justify the adverse scaling parameters in the
AARI#3 versions, which have been used by the authors for further publications for some years
since they were developed in 2006. By stating that they use version AARI#3 scaling parameters for
calculations of PC indices and then demonstrate in their Figs. 6, 7, and 8 small differences between
PC index values derived by using scaling parameters of two slightly different AARI_1995-2005
(AARI#4) versions they have overlooked the potential failure of the AARI_1998-2001 (AARI#3)
version from 2006.
- Contrary to statements in the text, the AARI#3 scaling parameters have been replaced in their Fig.
1 by those of the more recent version, AARI_1995-2005 (AARI#4) for the "with QDC" parameters.
The "without QDC" parameters have a different, indefinable basis and have obviously incorrect
relations to the "with QDC" parameters. With the mingling of parameter versions, the investigations
relating to the use of quiet day levels ("with QDC") or just base levels ("without QDC") for the
reference levels in the processing of polar magnetic data for derivation of PC index values have
generated the incorrect results reported in their Figs. 2 and 3.
- The small differences between PC index values derived by using scaling parameters from two
slightly different, but otherwise indefinable AARI#4 versions have been used to postulate
invariability with respect to the solar cycle of derived index scaling parameters in the title, abstract
and conclusions of the TPJ2011 publication. On the presented basis, this postulate must be
considered unsubstantiated.



**Data availability**
Geomagnetic data from Vostok were supplied from the INTERMAGNET data service web portal at
http://intermagnet.org.
Solar wind plasma and magnetic field data based on data from the ACE, IMP, GeoTail, and WIND
space missions were supplied from the OMNIweb data service at http://omniweb.gsfc.nasa.gov .
DMI PCN and PCS derivation methods and scaling parameters used since 2006 in PC index
publications issued from DMI are documented in DMI Scientific Report, SR-06-04 from 2006
(revised 2007) available at http://www.dmi.dk/fileadmin/Rapporter/SR/sr06-04.pdf .
This report was updated in 2016 to use the same data from epoch 1998 to 2009 as those used for the
IAGA-endorsed PC index version while the methodology has remained the same. The report is
available at https://www.dmi.dk/fileadmin/user_upload/Rapporter/TR/2016/SR-16-22-PCindex.pdf
Concerning files of scaling parameter values corresponding accurately to the colour-coded displays
in Troshichev et al. (2006, 2011) and precise values of the reference quiet day variations (QDCs),
requests should be directed to Drs. O. A. Troshichev and A. S. Janzhura at the Arctic and Antarctic
Research Institute in St. Petersburg, Russia.
Tables of the PCS scaling parameter values read from the colour-coded diagrams in Troshichev et
al., 2006 are provided in Table A1 of the Appendix. Tables of hourly mean values of the scaling
coefficients from AARI files (Parameters2011.rar, Janzhura 21-06-2011), for epoch 1995-2005 are
included in Table A2 of the Appendix.
**Acknowledgments.** The staffs at the observatories in Qaanaaq and Vostok and their supporting
institutes, the Danish Meteorological Institute, the Danish Space Research Institute (DTU Space),
and the Arctic and Antarctic Research Institute in St. Petersburg, Russia, are gratefully
acknowledged for providing high-quality geomagnetic data for this study. The efficient provision of
geomagnetic data from the INTERMAGNET data service centre, the supply of data from IMP,
WIND, ACE, and GeoTail missions and the excellent performance of the OMNIweb data portals
are greatly appreciated. The author gratefully acknowledges the collaboration and many rewarding
discussions in the past with Drs. O. A. Troshichev and A. S. Janzhura at the Arctic and Antarctic
Research Institute in St. Petersburg, Russia.

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

## Appendix.

**Scaling parameter values.**

**Table A1**. Hourly mean values of PCS Scaling coefficients read from Fig. 3 of Troshichev et al. (2006)

```
PCS Optimum angle parameters (in deg.) based on Vostok data 1998-2001.
HR   JAN   FEB   MAR   APR   MAY   JUN   JUL   AUG   SEP   OCT   NOV   DEC
00   16.0  18.2  30.0  38.0  46.6  43.5  46.6  41.1  39.8  30.6  21.7  16.0
01    8.8  13.0  26.5  37.0  48.0  43.5  46.4  37.2  36.4  26.2  17.0  11.0
02    1.5   7.4  23.5  36.5  49.5  44.6  45.6  36.0  33.5  22.0  12.2   6.5
03   -6.0   2.6  22.0  36.7  50.0  48.8  45.4  37.0  32.6  20.8   9.0   4.0
```



```
639  04 -10.2   0.6  21.6  37.8  50.5  54.0  48.0  41.0  33.0  21.4   9.3   3.2
640  05 -11.0   1.3  23.8  41.6  54.0  59.5  54.0  48.2  36.8  23.6  13.0   5.0
641  06  -6.6   4.0  29.4  45.7  57.5  64.0  60.4  55.0  42.0  27.2  17.5  10.2
642  07   2.0  11.5  36.0  50.2  61.2  67.0  66.4  61.0  47.0  32.2  23.2  16.0
643  08  12.0  18.6  41.3  54.4  62.4  66.2  67.4  65.2  52.8  39.0  29.0  21.0
644  09  20.5  26.4  45.3  56.8  62.2  63.3  66.8  66.7  58.0  46.0  34.0  25.0
645  10  26.6  33.0  48.6  58.0  61.0  59.0  64.2  65.5  61.2  50.2  38.0  27.5
646  11  30.8  38.2  52.0  58.0  58.5  53.3  58.8  63.2  64.0  54.2  43.0  31.0
647  12  34.7  42.5  54.2  57.8  55.5  49.6  52.0  59.4  65.8  59.0  47.5  35.0
648  13  39.0  46.0  54.4  58.0  52.8  47.0  46.8  56.4  66.6  64.2  52.5  40.4
649  14  44.8  50.4  54.4  57.3  49.8  45.2  45.2  55.5  65.8  67.0  57.3  46.5
650  15  50.8  54.4  54.5  54.6  47.5  45.0  45.6  55.2  64.5  68.6  61.2  51.6
651  16  53.7  56.6  54.5  52.7  46.0  46.2  46.0  55.0  62.8  69.2  63.0  56.8
652  17  53.8  56.5  54.4  51.0  46.0  47.7  46.0  54.7  60.8  68.8  61.8  57.4
653  18  50.3  54.2  52.6  49.3  46.4  48.6  45.7  54.0  58.8  67.0  58.5  54.6
654  19  45.5  49.2  49.0  47.4  46.8  49.0  45.6  53.0  56.4  64.0  53.5  48.8
655  20  41.0  41.7  44.8  45.8  46.6  49.0  45.8  51.3  53.8  59.5  47.6  41.0
656  21  35.8  35.8  39.7  43.2  46.2  48.3  46.4  49.3  51.6  53.7  41.2  33.0
657  22  30.5  30.0  36.0  41.0  46.0  47.2  46.6  47.3  48.4  47.2  35.0  26.8
658  23  24.0  24.7  32.8  39.4  46.2  46.0  46.6  44.8  44.6  39.2  27.8  20.8
659
PCS Slope values (in nT/(mV/m)) based on Vostok data 1998-2001.
HR   JAN   FEB   MAR   APR   MAY   JUN   JUL   AUG   SEP   OCT   NOV   DEC
662  00  47.0  44.5  41.5  38.5  37.5  37.5  38.5  40.5  43.5  45.5  48.0  49.0
663  01  47.5  44.5  41.5  38.5  37.0  37.0  38.5  40.5  43.5  46.0  48.5  49.0
664  02  47.5  45.0  41.5  38.5  36.5  36.5  37.5  39.5  42.5  45.5  48.0  48.5
665  03  47.0  45.0  41.5  38.5  36.5  36.0  36.5  38.5  41.5  44.5  47.0  48.0
666  04  45.5  44.5  41.5  37.5  35.0  33.5  33.5  35.5  39.5  42.5  46.0  46.5
667  05  46.5  45.5  42.5  37.5  34.5  32.5  32.5  34.5  39.5  43.0  45.5  47.0
668  06  44.0  43.0  40.5  36.0  33.0  31.5  32.0  34.5  39.0  42.5  45.0  45.5
669  07  43.0  41.5  38.5  34.5  32.0  31.0  32.5  35.0  39.5  43.5  45.0  45.0
670  08  43.0  41.5  38.5  34.5  32.5  32.0  33.5  36.5  40.5  44.5  45.5  45.5
671  09  43.5  41.5  38.0  34.5  32.5  32.5  34.0  37.5  42.0  45.0  45.0  46.0
672  10  43.0  41.5  38.5  35.5  32.5  32.0  33.0  35.5  39.5  43.0  44.5  44.5
673  11  43.0  42.0  39.5  36.0  33.0  31.5  31.5  33.5  37.5  41.5  43.5  43.5
674  12  43.0  42.0  40.0  36.0  32.5  30.5  30.5  32.0  35.5  40.0  42.5  43.5
675  13  44.0  42.5  40.5  36.5  32.5  30.5  29.5  31.5  35.5  39.5  43.0  44.5
676  14  43.0  42.0  39.5  35.5  31.5  29.5  29.0  31.0  34.5  38.5  42.5  43.5
677  15  41.0  40.0  37.5  34.0  31.0  29.5  29.5  31.0  33.5  37.5  40.5  41.5
678  16  38.5  36.5  34.5  32.5  30.5  29.0  29.5  31.0  33.0  35.5  38.5  39.0
679  17  38.0  36.5  35.0  32.5  30.5  29.0  29.5  30.5  33.0  35.5  37.5  38.5
680  18  38.5  37.0  35.5  33.5  31.0  30.0  30.5  31.5  34.0  36.5  38.5  39.5
681  19  40.5  39.0  37.5  35.5  33.0  31.5  31.5  32.5  35.0  37.5  40.0  40.5
682  20  43.5  42.5  40.5  38.0  35.5  34.0  34.5  35.5  38.5  40.5  43.5  44.0
683  21  45.5  44.5  42.5  39.5  37.0  36.0  36.5  38.0  40.5  43.5  46.5  46.5
684  22  47.5  45.5  43.0  40.5  38.0  37.0  38.0  40.0  42.5  45.5  48.5  48.5
685  23  47.0  44.5  41.5  39.0  37.5  37.0  38.5  40.5  43.5  46.5  48.5  49.0
PCS Intercept values (in nT) based on Vostok data 1998-2001.
HR   JAN   FEB   MAR   APR   MAY   JUN   JUL   AUG   SEP   OCT   NOV   DEC
689  00  -4.0  -4.0  -4.0  -3.0  -3.0  -3.0  -2.0  -3.0  -4.0  -5.0  -5.0  -5.0
690  01  -3.0  -3.0  -3.0  -2.0  -2.0  -2.0  -1.0  -1.0  -2.0  -4.0  -4.0  -4.0
691  02  -3.0  -4.0  -4.0  -3.0  -2.0  -2.0   0.0   0.0  -1.0  -3.0  -3.0  -3.0
692  03  -4.0  -5.0  -6.0  -4.0  -3.0  -2.0   0.0   1.0  -1.0  -3.0  -3.0  -4.0
693  04  -7.0  -9.0  -9.0  -6.0  -4.0  -1.0   2.0   2.0  -1.0  -4.0  -5.0  -6.0
694  05 -14.0 -15.0 -14.0  -9.0  -5.0  -1.0   2.0   1.0  -4.0  -8.0 -11.0 -12.0
695  06 -16.0 -17.0 -15.0 -10.0  -5.0  -1.0   1.0  -1.0  -7.0 -12.0 -15.0 -15.0
696  07 -17.0 -17.0 -15.0 -10.0  -6.0  -2.0  -1.0  -3.0 -10.0 -15.0 -17.0 -17.0
697  08 -17.0 -17.0 -15.0 -11.0  -6.0  -4.0  -3.0  -6.0 -11.0 -16.0 -18.0 -18.0
```





```
698    09 -16.0 -15.0 -13.0 -10.0  -6.0  -5.0  -5.0  -7.0 -12.0 -16.0 -17.0 -17.0
699    10 -13.0 -13.0 -12.0 -10.0  -7.0  -5.0  -5.0  -6.0 -10.0 -12.0 -13.0 -13.0
700    11 -14.0 -14.0 -13.0 -11.0  -8.0  -6.0  -5.0  -6.0  -9.0 -11.0 -12.0 -13.0
701    12 -15.0 -16.0 -15.0 -12.0  -9.0  -7.0  -5.0  -6.0  -8.0 -11.0 -13.0 -14.0
702    13 -17.0 -18.0 -17.0 -15.0 -11.0  -8.0  -6.0  -7.0  -9.0 -12.0 -15.0 -16.0
703    14 -17.0 -18.0 -17.0 -15.0 -11.0  -8.0  -7.0  -7.0  -9.0 -11.0 -14.0 -15.0
704    15 -14.0 -15.0 -14.0 -13.0 -11.0  -8.0  -7.0  -7.0  -8.0 -10.0 -11.0 -13.0
705    16 -11.0 -11.0 -12.0 -11.0 -10.0  -8.0  -8.0  -7.0  -7.0  -8.0  -9.0  -9.0
706    17  -9.0 -10.0 -11.0 -11.0 -10.0  -8.0  -7.0  -7.0  -7.0  -8.0  -8.0  -8.0
707    18  -9.0  -9.0 -10.0 -10.0  -9.0  -8.0  -7.0  -7.0  -8.0  -8.0  -9.0  -9.0
708    19  -9.0 -10.0 -11.0 -11.0 -10.0  -8.0  -7.0  -7.0  -8.0  -8.0  -9.0  -9.0
709    20 -11.0 -12.0 -13.0 -12.0 -10.0  -9.0  -8.0  -8.0  -9.0 -10.0 -11.0 -11.0
710    21 -12.0 -13.0 -13.0 -12.0 -10.0  -9.0  -8.0  -8.0 -10.0 -11.0 -12.0 -12.0
711    22 -11.0 -11.0 -11.0 -10.0  -8.0  -7.0  -7.0  -7.0 -10.0 -11.0 -12.0 -12.0
712    23  -8.0  -7.0  -7.0  -6.0  -5.0  -5.0  -5.0  -5.0  -7.0  -9.0  -9.0  -9.0
```

**Table A2.** Hourly mean values of PCS Scaling coefficients from AARI file (Parameters2011.rar, 21-06-
2011)

```
AARI PCS Optimum angle values (deg.) based on Vostok data 1995-2005. Angle_Fi.1M
HR   JAN   FEB   MAR   APR   MAY   JUN   JUL   AUG   SEP   OCT   NOV   DEC
718     0  44.8  51.3  59.7  66.5  69.0  67.3  62.8  57.2  51.5  46.4  42.7  41.4
719     1  39.4  46.9  56.8  65.4  69.2  68.0  63.4  57.1  50.4  44.1  39.2  37.0
720     2  34.5  42.4  53.3  63.2  68.1  67.5  62.9  56.3  48.8  41.3  35.3  32.3
721     3  30.3  38.6  50.4  61.2  67.2  67.3  62.9  56.0  48.1  39.8  32.8  28.9
722     4  27.3  35.9  48.2  59.9  66.6  67.3  63.2  56.4  48.3  39.4  31.5  26.6
723     5  26.0  34.4  47.0  59.1  66.4  67.5  63.9  57.5  49.5  40.3  31.8  26.2
724     6  26.9  34.9  47.3  59.5  67.0  68.4  65.3  59.3  51.7  42.7  34.0  28.0
725     7  30.3  37.7  49.4  61.0  68.2  69.7  67.0  61.7  54.6  46.1  37.7  31.7
726     8  35.0  41.6  52.3  62.8  69.2  70.5  68.3  63.9  57.8  50.2  42.4  36.7
727     9  40.1  46.0  55.5  64.6  69.8  70.4  68.6  65.3  60.6  54.3  47.4  42.0
728    10  44.8  50.4  58.9  66.5  69.9  69.4  67.5  65.5  62.7  58.0  51.9  46.7
729    11  48.7  54.2  61.9  67.9  69.1  67.2  65.2  64.6  64.0  61.0  55.5  50.4
730    12  52.7  57.9  64.6  68.6  67.9  64.6  62.7  63.5  64.9  63.5  58.8  54.0
731    13  57.3  61.9  67.1  69.1  66.7  62.4  60.5  62.3  65.4  65.8  62.2  58.0
732    14  62.1  65.8  69.2  69.2  65.4  60.7  58.9  61.4  65.7  67.6  65.5  62.3
733    15  66.2  68.9  70.5  68.9  64.4  59.8  58.1  60.8  65.7  68.8  68.2  66.2
734    16  69.2  71.0  71.3  68.6  63.9  59.7  58.2  60.6  65.4  69.1  69.8  68.9
735    17  70.5  71.8  71.4  68.4  63.9  60.1  58.5  60.3  64.6  68.4  69.8  69.7
736    18  69.8  71.3  71.0  68.2  64.2  60.6  58.9  60.0  63.4  66.9  68.4  68.6
737    19  68.0  69.9  70.3  68.3  64.9  61.5  59.4  59.6  61.9  64.5  65.8  66.1
738    20  65.3  68.0  69.5  68.6  65.9  62.8  60.2  59.2  60.1  61.5  62.4  62.9
739    21  61.7  65.2  68.1  68.7  67.2  64.3  61.1  59.0  58.4  58.5  58.7  59.0
740    22  57.5  62.0  66.5  68.9  68.5  66.0  62.2  58.8  56.7  55.5  54.8  54.8
741    23  51.5  56.9  63.2  67.8  68.8  66.6  62.4  57.7  53.8  50.6  48.4  48.0
AARI PCS Slope values  (nT/(mV/m))  based on Vostok data 1995-2005. Coeff_alpha.1M
HR   JAN   FEB   MAR   APR   MAY   JUN   JUL   AUG   SEP   OCT   NOV   DEC
745     0  45.3  45.2  43.2  39.3  34.8  31.7  31.5  34.2  37.8  40.5  42.4  44.1
746     1  45.7  45.6  43.3  39.4  34.8  31.6  31.5  34.3  38.0  40.5  42.4  44.4
747     2  46.6  46.0  43.3  39.0  34.4  31.2  31.1  34.2  38.0  40.6  42.8  45.2
748     3  47.4  45.9  42.4  37.8  33.2  30.2  30.3  33.6  37.5  40.4  43.3  46.3
749     4  47.7  45.2  40.9  36.1  31.7  29.0  29.3  32.4  36.3  39.6  43.3  46.9
750     5  47.6  44.4  39.6  34.7  30.6  28.2  28.4  31.2  34.8  38.4  42.8  46.8
751     6  46.5  43.3  38.4  33.7  29.9  27.7  27.7  30.2  33.7  37.4  41.9  45.8
752     7  44.1  41.0  36.6  32.7  29.5  27.4  27.3  29.7  33.2  36.9  41.0  44.0
753     8  41.7  38.6  35.0  31.9  29.2  27.4  27.4  29.5  33.0  36.9  40.5  42.5
754     9  41.4  37.7  34.3  31.5  28.9  27.2  27.3  29.4  33.0  37.3  41.3  43.0
755    10  43.3  38.7  34.5  31.1  28.2  26.5  26.7  29.0  33.0  38.1  43.2  45.4
```



```
756  11  45.5  40.0  34.7  30.6  27.5  25.8  26.0  28.5  32.8  38.6  44.7  47.8
757  12  46.6  40.9  34.9  30.2  27.0  25.2  25.4  27.9  32.6  38.8  45.3  48.7
758  13  46.4  41.1  34.9  29.9  26.6  24.7  24.8  27.5  32.6  38.9  45.0  48.2
759  14  44.9  40.3  34.5  29.6  26.2  24.2  24.2  27.1  32.5  38.6  43.8  46.4
760  15  42.8  38.9  33.9  29.3  25.8  23.8  24.0  27.0  32.3  37.8  42.1  44.1
761  16  41.2  38.1  33.7  29.3  25.8  23.9  24.1  27.0  31.9  36.9  40.7  42.3
762  17  40.7  38.4  34.4  30.0  26.5  24.6  24.7  27.3  31.7  36.3  39.7  41.3
763  18  40.8  39.2  35.7  31.4  27.9  26.0  25.9  28.3  32.3  36.4  39.5  41.0
764  19  41.1  40.1  37.1  33.0  29.6  27.6  27.4  29.7  33.6  37.2  39.7  41.0
765  20  41.5  41.1  38.4  34.6  31.1  28.8  28.5  30.9  34.7  37.9  40.1  41.1
766  21  42.3  42.2  39.9  36.1  32.3  29.6  29.3  31.8  35.5  38.7  40.7  41.7
767  22  43.5  43.4  41.3  37.6  33.4  30.5  30.2  32.7  36.4  39.4  41.4  42.7
768  23  44.6  44.6  42.6  38.7  34.4  31.4  31.1  33.7  37.3  40.1  42.1  43.6
AARI PCS Intercept values (nT) based on Vostok data 1995-2005. Coeff_beta.1M
HR   JAN   FEB   MAR   APR   MAY   JUN   JUL   AUG   SEP   OCT   NOV   DEC
772   0   0.1  -1.4  -2.8  -3.5  -3.4  -3.1  -2.6  -1.8  -0.2   2.1   3.0   1.7
773   1   0.8  -0.8  -2.4  -3.3  -3.5  -3.2  -2.7  -1.8   0.2   2.9   4.0   2.6
774   2   0.8  -0.6  -2.2  -3.3  -3.7  -3.6  -3.0  -1.8   0.3   3.4   4.5   2.8
775   3   0.3  -0.8  -2.3  -3.5  -4.1  -4.1  -3.4  -2.0   0.3   3.4   4.4   2.4
776   4  -0.3  -1.4  -2.7  -3.9  -4.5  -4.5  -3.9  -2.5  -0.1   2.8   3.6   1.6
777   5  -1.0  -2.1  -3.4  -4.4  -4.9  -5.0  -4.5  -3.2  -1.0   1.7   2.3   0.7
778   6  -1.6  -2.7  -4.1  -5.0  -5.3  -5.4  -5.1  -4.0  -2.0   0.3   1.0  -0.2
779   7  -2.4  -3.5  -4.7  -5.5  -5.8  -5.8  -5.6  -4.8  -3.2  -1.4  -0.7  -1.4
780   8  -3.7  -4.4  -5.3  -6.0  -6.2  -6.3  -6.2  -5.7  -4.6  -3.4  -3.0  -3.2
781   9  -5.6  -5.5  -5.9  -6.4  -6.7  -6.8  -6.8  -6.6  -6.1  -5.6  -5.7  -5.7
782  10  -7.7  -6.8  -6.6  -6.9  -7.2  -7.2  -7.2  -7.4  -7.5  -7.7  -8.3  -8.4
783  11  -9.7  -8.2  -7.4  -7.4  -7.5  -7.5  -7.5  -8.0  -8.7  -9.6 -10.5 -10.7
784  12 -11.1  -9.4  -8.2  -7.8  -7.7  -7.6  -7.7  -8.5  -9.7 -10.8 -11.9 -12.1
785  13 -11.7 -10.2  -8.8  -8.1  -7.8  -7.6  -7.7  -8.7 -10.1 -11.3 -12.3 -12.5
786  14 -11.7 -10.5  -9.2  -8.3  -7.9  -7.6  -7.7  -8.6 -10.0 -11.1 -11.9 -12.2
787  15 -11.4 -10.4  -9.2  -8.3  -7.8  -7.6  -7.6  -8.4  -9.5 -10.4 -11.0 -11.5
788  16 -10.8 -10.1  -8.9  -8.0  -7.6  -7.4  -7.5  -8.0  -8.7  -9.2  -9.8 -10.5
789  17 -10.1  -9.7  -8.5  -7.5  -7.1  -7.0  -7.2  -7.5  -7.9  -8.1  -8.7  -9.6
790  18  -9.4  -9.2  -8.1  -7.0  -6.5  -6.5  -6.6  -6.8  -6.9  -7.1  -7.7  -8.7
791  19  -8.4  -8.4  -7.5  -6.5  -5.9  -5.7  -5.7  -5.7  -5.7  -5.9  -6.5  -7.6
792  20  -7.0  -7.3  -6.7  -6.0  -5.4  -5.0  -4.7  -4.5  -4.4  -4.4  -5.1  -6.1
793  21  -5.3  -5.9  -5.8  -5.4  -4.8  -4.3  -3.9  -3.5  -3.0  -2.8  -3.2  -4.2
794  22  -3.3  -4.3  -4.7  -4.7  -4.2  -3.7  -3.3  -2.7  -1.8  -0.9  -1.0  -2.1
795  23  -1.4  -2.6  -3.6  -4.0  -3.7  -3.2  -2.8  -2.1  -0.8   0.8   1.2   0.1

796

797
```