# Peer review of "Title: Comment on "Invariability of relationship between the polar cap magnetic activity and geoeffective interplanetary electric field" by Troshichev et al. (2011)."

_Annales Geophysicae, 2020_

## Referee Comment (RC1) · Anonymous Referee #1 · 7 Jan 2021

This manuscript comment [Stauning 2020] identifies a defect in a paper by [Troshichev et al 2006]. Specifically, it is shown that the original analysis appears to have mistakenly used interplanetary magnetic field (IMF) vector components in the GSE reference frame instead of GSM as intended. It is not entirely clear how this error was discovered ("New analyses has disclosed ...") and whether it has been confirmed by review of the original code. Figure 1 presents an interval of solar wind data By and Bz magnetic field components in GSE and GSM coordinates compared with a similar plot taken from the original paper. Although the magnetic field components in GSM and GSM are very

similar for much of this interval, they are distinctly different for several hours, during which it is clear that the the [TJS2006] solar wind plot is not of GSM but GSE instead.

[Stauning2020] notes that "The mistake had no strong impact on the remaining presentation of the PC index concept in [TJS2006]. Usually, such a mistake would not attract attention after the many years that have passed since the publishing in 2006. However, the incorrect feature drags a trail of erroneous relations and invalid statements presented in publications on polar cap indices issued since 2006 extending up to present (2020)."

This comment is primarily concerned with a study by [TPJ2011] which was thought to have inherited the GSE/GSM error from [TJS2006]. The goal of [TPJ2011] was to explore whether a single set of model coefficients could be utilized for all levels of solar activity. [Stauning2020] expected that any effect due to GSE/GSM errors might be evident as changes in the rotation angle of the local horizontal geomagntic field ($\phi$).

However, [Stauning2020] notes A problem for the analysis of possible effects of the invalid PCS scaling parameters derived in TJS2006 by using IMF components in their GSE representation is the unavailability of numerical files of the parameters. Instead, the colour-coded diagrams have been "manually" read-off to be converted to numerical files.

[Stauning2020] asserts that the results obtained by [TPJ2011] are in error and "It has not been possible to deduce the origin of the scaling parameters actually used for two PCS versions being compared in Figs. 6, 7, and 8 of TPJ2011". and "Furthermore, the "without QDC" curves are not derived from calculations of scaling parameters from the "with QDC" version just without using QDCs but are of indefinable origin".

[Stauning2020] presents a detailed comparison of several different parameter sets, concluding that the corresponding results in [TPJ2011] were inconsistent with the stated set of coefficients. Subsequent analysis leads to the conclusion that key results are partially due to erroneous quiet day curves (QDC).

Finally, [Stauning2020] returns to the GSE/GSM discrepancy and concludes that it could result in quasi-random error of 1-2 mV/m under quiet to disturbed conditions and 10 mV/m or more during storm times.

In my opinion, this manuscript contains several noteworthy results and should be published. As a comment it should serve as a useful caution regarding the conclusions of [TPJ2011]. It also provides an opportunity for the authors of [TPJ2011] to present a detailed reply about their methods and conclusions.

That still leaves the question of how to provide a persistent warning regarding [TJS2006]. Ideally, the authors of [TJS2006] could write a correction or erratum in Annales Geophysicae which would appear in a citation search. Alternatively [Stauning2020] could be used as the basis for a second comment very similar to this one.

---

## Short Comment (SC1) · 15 Jan 2021

Comments to MS https://doi.org/10.5194/angeo-2020-52 Title: Comment on "Invariability of relationship between the polar cap magnetic activity and geoeffective interplanetary electric field" by Troshichev et al. (2011) Author: Peter Stauning, Danish Meteorological Institute, Lyngbyvej 100, Copenhagen.

Author (Dr.Stauning) "encloses" that parameters $\alpha$, $\beta$ and $\varphi$ presented in [Troshichev et al., 2006] were derived with use of the GSE (Geocentric Solar Ecliptic) representation instead of the GSM (Geocentric Solar Magnetosphere) representation. Basing on this "discovery" he makes the conclusion that the unified PC method described in [Troshichev et al., 2006] is invalid and this mistake has caused a trail of incorrect relations and wrong conclusions extending since 2006 up to now (2020). In reality, the paper [Troshichev et al., 2006] describes the main principles and procedures used in the unified PC index derivation method. Just this method was approved as the best method by the IAGA Division V-DAT at special meeting in Vienna in May 2010. Before the meeting a special Task Force team fulfilled the comprehensive analysis of three competitive methods: DMI official, Dr.Stauning private, and AARI method (see [Mc-Creadie and Menvielle, 2010]). The new PC index was endorsed by IAGA as a new index of magnetic activity basing on the IAGA Division V-DAT recommendation. The paper [Troshichev et al., 2006] included also some figures with aim to illustrate the proposed procedures. In 2009 Dr. Stauning found that illustrations were made with use of the GSE instead of GSM. As a result, all parameters $\alpha$, $\beta$ and $\varphi$ were recalculated with use of GSM, and just these parameters were used in all subsequent analyses. As this takes place the main principles and procedures put forward in [Troshichev et al., 2006] remained unchanged. Since parameters $\alpha$, $\beta$ and $\varphi$ obtained with use of GSE or GSM were not principally distinct, the Figures in the paper remained without changes. Dr. Stauning was informed about this situation in 2009. It should be particularly emphasized that the PC index is calculated at present with use of GSM derived parameters and Dr. Stauning perfectly knows about it. Nevertheless, he calls into question the approved method with referring to old illustrations. The validity of the unified PC derivation method has been perfectly testified by its close relationships with magnetospheric disturbances. At present PC index is very popular: according to Research Gate above 1500 persons read every year publications on the IAGA endorsed PC index. Taking into account all these circumstances my conclusion is the following: the Dr.Stauning's papers do not have scientific value and are not worthy of publication.

---

## Author Comment (AC1) · 16 Jan 2021

Copenhagen 16 January 2021/PSt

Reply to Interactive comments by Anonymous Referee #1 on "Comment on "Invariability of relationship between the polar cap magnetic activity and geoeffective interplanetary electric field" by Troshichev et al. (2011)" by Peter Stauning.

The comments by the Anonymous Referee #1 expresses much the same concerns as the commentary note over the mistake in using the interplanetary magnetic field

components IMF By and Bz in their GSE version instead of the prescribed GSM version and also suggests issuing a "persistent warning regarding TJS2006". As co-author of Troshichev et al. (2006) I share the responsibility for the error. I have some time ago without success suggested to Dr. Troshichev, first and corresponding author of the publication, to issue a corrigendum note to J. Geophys. Res. Space Physics where the article was published. Actually, I have also discovered errors in the calculation code used for the present calculations of the scaling parameters for the "definitive" PCN index values made at the Danish Space Research Institute, DTU Space, co-responsible with AARI for the IAGA-endorsed PC indices. I have informed AARI as well as DTU Space (and IAGA officers) of the supposed failure but received little response. Thus I have submitted a manuscript with description of the errors, which affect the IAGA-endorsed version as well as previous PC index versions issued from AARI. The manuscript is presently in review.

As part of the manuscript discussing PC index versions developed at AARI I have suggested to issue a corrigendum note for the mistake on the GSE representation of IMF components in Troshichev et al. (2006). The suggested text is: "The publication, Troshichev, Janzhura, and Stauning (2006) by mistake used the Interplanetary Magnetic Field (IMF) components BY and BZ in the geocentric solar ecliptic (GSE) system instead of the geocentric solar magnetospheric (GSM) system in the calculation of PC index scaling parameters. The incorrect parameter sets are displayed in the colour-coded diagrams in Fig. 3 of the publication. The remaining part of the article is not much affected by the incorrect scaling parameters. However, the parameter sets, now named AARI#3 versions, based on data from epoch 1998-2001, have been used in further publications issued between 2006 and 2011. Thus, we should caution against uncritical use of relations and conclusions published in papers that may have used the invalid AARI#3 versions of scaling parameters and derived PCN and PCS index values".

In conclusion, the best approach would be a corrigendum note to Troshichev et al.

(2006) issued jointly by the three co-authors like we did with the previous corrigendum note published in Troshichev, Janzhura, and Stauning (2009). The above corrigendum text submitted to J. Geophys. Res. Space Physics is just an escape solution to rectify the mistake made in 2006 and to caution against its possible consequences. The commentary notes on Troshichev et al. (2011) [TPJ2011] and also the comment on Janzhura and Troshichev (2011) [JT2011] submitted for review at the AnGeo Interactive Discussion portal are about publications issued by Annales Geophysicae and should in my opinion be kept there. Furthermore, as noted by Referee #1 concerning TPJ2011 and equally valid for JT2011, it "provides an opportunity for the authors of TPJ2011 to present a detailed reply about their methods and conclusions".

Copenhagen 16 January 2021

Peter Stauning

---

## Author Comment (AC2) · 16 Jan 2021

Copenhagen 16 January 2021/PSt

Reply to Interactive comments by Dr. O.A. Troshichev on "Comment on "Invariability of relationship between the polar cap magnetic activity and geoeffective interplanetary electric field" by Troshichev et al. (2011)" by Peter Stauning.

It may be noted that the comments by Dr. Troshichev confirm that the derivation in Troshichev et al. (2006) of the scaling parameters in the version AARI_1998-2001

(i.e. AARI#3 according to McCready and Menvielle, 2010) was based on using the interplanetary magnetic field components IMF By and Bz in their GSE representation instead of the prescribed GSM version. This unfortunate feature has never been published before in spite of many references to the publication and use of its scaling parameter illustrations, most recent in Figs. 2.3 and 2.9 of Troshichev (2017: Polar Cap magnetic activity (PC index) and space weather monitoring, ISBN: 978-3-8381-8012-0. On top, this publication holds in its Fig. 9.2 a reproduction of the IMF By and Bz components from Fig. 7 of Troshichev et al. (2006) (also presented in Fig. 1a of the commentary) without mentioning that it presents the GSE and not the GSM version.

However, the main problem dealt with in the commentary is the stated use in Troshichev et al. (2011) of the "solar max" scaling parameters in the version, AARI_1998-2001 (AARI#3), for the analyses documented in Figs. 1, 2, 3, 6, 7, and 8 while in reality other versions were used as basis for the analyses and conclusions.

The commentary illustrates the considerable differences between the displays in Figs. 1, 2, 3, 6, 7, an 8 of Troshichev et al. (2011) and the corresponding displays resulting from really using the AARI_1998-2001 (AARI#3) version in the analyses. The commentary also display in its Fig. 15 the at times quite large differences between PC index values calculated using the AARI_1998-2001 (AARI#3) scaling parameter version and the corresponding PC index values calculated by using more recent scaling parameter values. Thus, the absence of objections from Dr. Troshichev against the presented material justifies the commentary and verifies that the analyses presented in Troshichev et al. (2011) are incorrect and the conclusions, therefore, are unsubstantiated.

Copenhagen 16 January 2021

Peter Stauning

---

## Author Comment (AC3) · 27 Jan 2021

Copenhagen 27 January 2021/PSt

Final Author Comments to ANGEOD discussions on angeo-2020-52

(1) Summary of submitted contribution: P. Stauning: "Comment on "Invariability of relationship between the polar cap magnetic activity and geoeffective interplanetary electric field" by Troshichev et al. (2011)." Angeo-2020-52.

The main issue in the commentary is the incorrect reference to the PC index scaling parameter version "AARI_1998-2001" (AARI#3) in statements and illustrations where different scaling parameters are actually used. The scaling parameters AARI_1998-2001 based on data from the solar maximum years 1998-2001 were derived in Troshichev et al. (2006), presented in their Fig. 3, and named AARI#3 version by McCready and Menvielle (2010) in their Table 1.

Unfortunately, the Interplanetary Magnetic Field (IMF) component IMF BY and IMF BZ in the Geocentric Solar Ecliptic (GSE) representation were used instead of the prescribed Geocentric Solar Magnetospheric (GSM) representation for the geoeffective interplanetary electric field applied for the derivation of the index scaling parameters. Dr. Troshichev was made aware of the mistake in 2009.

In the publication commented here, Troshichev et al. (2011), the authors state that they have used scaling parameters of the (GSE-based) AARI#3 PC index version from 2006 in their work but omit to report that they have actually substituted parameters from a more recent GSM-based version instead. In my view, deliberately using data different from the announced data basis is not ac ceptable and should be reported.

The main aim for Troshichev et al. (2011) is to demonstrate that using index scaling parameters derived from an epoch of high solar activity (1998-2001) would generate PC index values close to values generated by using index scaling parameters derived from an epoch of low solar activity (1997+2007-2009). Thus, they conclude on basis of results obtained with a GSM-based scaling parameter version substituted for the AARI#3 version from Troshichev et al. (2006) that their version of index scaling parameters are invariant over the solar cycle – that is – forever valid. By substituting a GSM-based AARI version for the GSE-based AARI#3 version the authors avoid disclosing the large differences in scaling parameters and derived PC index values in their results and illustrations that would result from the use of the announced AARI#3 version from Troshichev et al. (2006).

These differences are illustrated in the submitted commentary to document that the mingling of PC index versions have resulted in erroneous illustrations in Figs. 1, 2, 3, 6, 7, and 8 of Troshichev et al. (2011) and the issuing of a non-substantiated statement on the invariance of scaling parameters over the solar cycle. This statement is repeated in p. 73-74 of Troshichev and Janzhura (2012), in p. 340 of Troshichev (2012), in p. 15 of Troshichev (2017), and in p. 4 of ISO Report TR23989 with reference to the publication by Troshichev et al. (2011) commented here.

(2) Summary of Interactive Comments angeo-2020-52-RC1.pdf and Reply angeo-2020-52-AC1.

The reviewer asks in his contribution from 7 January whether the error in using GSE instead of GSM interplanetary magnetic field (IMF) vector components have been confirmed. This question is answered by Dr. Troshichev in his interactive comment in angeo-2020-52.SC1 from 15 January discussed below where he admits the error.

The reviewer concludes that "this manuscript contains several noteworthy results and should be published. As a comment it should serve as a useful caution regarding the conclusions of [TPJ2011]. It also provides an opportunity for the authors of [TPJ2011] to present a detailed reply about their methods and conclusions". This opportunity was not used by Dr. Troshichev in his interactive contribution from 15 January discussed below.

The reviewer mentions "that still leaves the question of how to provide a persistent warning regarding [TJS2006]. Ideally, the authors of [TJS2006] could write a correction or erratum in Annales Geophysicae which would appear in a citation search. Alternatively [Stauning2020] could be used as the basis for a second comment very similar to this one".

In my opinion an erratum or corrigendum should also appear in JGR Space Physics since the contribution by Troshichev et al. (2006) was published there. The submission by Stauning (2020), which is in review by JGR describes the GSE/GSM mistake

and suggests a text for a corrigendum note. The initial version of the submission was forwarded to the Editorial Board of Annales Geophysicae on 1 May 2020 and acknowledged by their reply on 13 May.

(3) Summary of Interactive Comments angeo-2020-52-SC1.pdf and Reply angeo-2020-52-AC2.

Dr. Troshichev mentions that "In 2009 Dr. Stauning found that illustrations were made with use of the GSE instead of the GSM. As a result, all parameters $\alpha$, $\beta$, and $\varphi$ were recalculated with use of GSM, and just these parameters were used in subsequent analyses".

This is the first public admission from Dr. Troshichev of the mistake made in Troshichev et al. (2006). It is possibly correct that GSM-based scaling parameter versions have been used by AARI after 2009. However, the basic problem in Troshichev et al. (2011) is the use of GSM-based scaling parameters while in the text and illustrations reference is made to the GSE-based scaling parameters from Troshichev et al. (2006). Thus, instead of admitting the GSE mistake, the publication by Troshichev et al. (2011), as it seems, attempts disguising the invalid scaling parameters set from 2006 by substituting another scaling parameter set.

In his interactive comment, angeo-2020-52-SC1.pdf, Dr. Troshichev argues that the method described in Troshichev et al. (2006) "was approved as the best method by the IAGA Division V-DAT at a special meeting in Vienna in May 2010". This argument is not seen to be relevant for the present discussion and the statement, furthermore, is incorrect. According to IAGA V-DAT there was no formal (business) V-DAT meeting in 2010 as such meetings are held during IAGA Assemblies, e.g., in 2009 or 2011. Furthermore, there is no IAGA documentation from a V-DAT meeting in May 2010.

In his interactive comment, Dr. Troshichev repels publishing the submitted commentary and avoids admitting that the substitution of another scaling parameter version could be a mistake. However, he forward no argument against the main theme of the

commentary stated in its abstract: "For the publication commented here, Troshichev et al. (2011), the authors state that they have used scaling parameters of the AARI#3 PC index version from 2006 in their work but they have actually substituted parameters from a more recent AARI_1995-2005 (AARI#4) version instead. The mingling of PC index versions have resulted in erroneous illustrations in their Figs. 1, 2, 3, 6, 7, and 8 and the issuing of non-substantiated statements".

Copenhagen 27 January 2021

Peter Stauning pst@dmi.dk